# Decision aids in patients with osteoporosis: A scoping review

**Yanyu Fang[1], Qin Jia [2]\*, Yaqin Dai[2], Siqi Li[1]**

**1** School of Nursing, Zhejiang Chinese Medical University, Hangzhou, Zhejiang Province, China,
**2** Department of Orthopedics, Zhejiang Provincial People's Hospital (People's Hospital Affiliated to Hangzhou Medical College), Hangzhou, Zhejiang Province, China

\* jq1227@163.com

## Abstract

### Introduction

Although various methods exist for osteoporosis prevention, most patients fail to receive optimal treatment due to information asymmetry between physicians and patients, as well as limited consultation time. Current literatures suggest that decision aids can support clinical decision-making by improving patients' risk perception and treatment acceptance.

### Objective

This scoping review described the use and effectiveness of decision aids in clinical decision-making among individuals with osteoporosis.

### Inclusion criteria

The review will include studies conducted in various countries that focus on decision-aiding interventions for people with osteoporosis in different settings and are published in English or Chinese.

### Methods

PubMed, CINAHL, Web of Science, Embase, Cochrane Library, China Knowledge Network, Wanfang Database, and China Biomedical Literature Database were searched. The search timeframe was from the establishment of the database to June 30, 2024. Studies that meet the inclusion criteria will be eligible for selection. The process of selecting eligible studies will then be summarized on a PRISMA-ScR chart. Collated in data-extraction tables will be authorship information, publication date, country, study site, sample information, study type, intervention form, content elements, application scope, and outcome indicators. The content elements, application

**Data availability statement:** All relevant data are within the manuscript and its Supporting information files.

**Funding:** This work was supported by the Zhejiang Provincial Department of Science and Technology Basic Public Welfare Research Funding Project (Grant number LTGY23H170005); the General project of Zhejiang Provincial Medical and Health Science and Technology Program (Grant number 2024KY644).

**Competing interests:** The authors have declared that no competing interests exist.

scope, and outcome indicators will be analyzed using a thematic analysis and summarized using a narrative summary.

## Conclusion

With strong efficacy and viability, DA greatly enhances patients' decision-making experience and decision quality. In order to provide patients with osteoporosis with high-quality decision-making support, it will be necessary to conduct large-scale, randomized controlled studies in the future with the goal of guaranteeing homogeneous interventions, expand the scope and meaning of the application of DA in osteoporosis, improve professional support during the decision-making process, create scientific and useful decision-making aids, and take specific actions.

## Introduction

Osteoporosis (OP) is a systemic skeletal disease characterized by low bone mass and deterioration of bone tissue microarchitecture, which leads to increased bone fragility and fracture susceptibility [1]. With a frequency of 32.0% among those over 65 in our nation [2], OP has become one of the main conditions preventing healthy aging. Treatment decisions are closely linked to significant patient morbidity and mortality, and while clinical practice guidelines support the diagnosis and management of OP [3], patients typically lack the time and information necessary to make timely decisions that align with their own values and preferences during the decision-making process of identifying the decision, gathering information, and weighing the risk-benefit.[4,5]. Patients urgently need clinical tools to help them make decisions about their living and treatment arrangements, as 85% of patients who require preventive treatment for OP either fail to select appropriate treatment options, have difficulty making decisions, or regret their decisions [6,7].

Decision Aids (DA) are decision support tools that, through brochures, videos, or web-based platforms, offer evidence-based information about options and outcomes related to an individual's health condition. They also help patients and healthcare professionals make decisions that align with their values and preferences by outlining the advantages and disadvantages of each option [8]. According to studies, DA can improve the accuracy of risk assessment [5], decrease decision-making conflicts [9], raise patient understanding and decision-making engagement [2], and boost the adoption of preventive interventions [6].

'High quality' DA can assist patients in making treatment decisions, according to the UK National Institute for Health and Treatment Excellence (NICE) recommendations [10]. Themes, content components, outcome indicator kinds, and the impact of DA on clinical decision-making in osteoporosis patients are, nevertheless, ambiguous and extremely varied. In summary, this scoping review systematically synthesizes decision topics, content elements, outcome types, and effectiveness of decision aids (DAs) in clinical decision-making for osteoporosis patients. By mapping existing evidence, we identify current knowledge gaps and provide actionable recommendations to guide future research directions in this field.

## Methods

We reported this review according to the Scoping Review extension of the Preferred Reporting Items for Systematic Review and Meta-Analysis statement (PRISMA-ScR) [11].

### Define the research question

Research questions: ① What are the main decision-making themes among osteoporosis patients that DA focuses on? ② What are the content elements through which DA provides decision support for people with osteoporosis? ③ How effective and acceptable is DA in decision-making among osteoporosis patients?

### Inclusion and exclusion criteria

The literature inclusion criteria were determined according to the PCC principle. ① Participants (P): patients meeting the diagnostic criteria for osteoporosis published by WHO [12] with T-value ≤ −2.5 for bone density of the midshaft bone (lumbar vertebrae 1–4, femoral neck, or total hip) or bone density of the distal 1/3 of the radius ≤ −2.5. ② Concept (C): decision-aided interventions for decision-making for patients with osteoporosis through DA. ③ Context (C)：The place where DA is applied, such as communities, nursing institutions or hospitals. Exclusion criteria: ①Relevant studies on cross-cultural debugging of decision aid tools; ② Conference papers, research protocols, non-Chinese and English literature; ③ Literature for which full text is not available.

### Search strategy

Search PubMed, CINAHL, Web of Science, Embase, Cochrane Library, China Knowledge Network, Wanfang Database, and China Biomedical Literature Database, and develop a search strategy by combining medical subject terms and keywords, and the English database was searched with PubMed as an example of search formula:

#1 ("Osteoporosis"[MeSH Terms] OR "osteoporos*"[Title/Abstract] OR "post traumatic osteoporos*"[Title/Abstract] OR "senile osteoporos*"[Title/ Abstract] OR "age related bone loss"[Title/Abstract] OR "age related bone losses"[Title/ Abstract] OR "age related osteoporos*"[Title/Abstract])

#2 ("Decision Support Techniques"[MeSH Terms] OR "decision aid*"[Title/Abstract] OR "decision support*"[Title/Abstract] OR "decision technolog*"[Title/Abstract] OR "decision technique*"[Title/Abstract] OR "decision algorithm*"[Title/ Abstract] OR "decision intervention*"[Title/ Abstract] OR "decision material"[Title/Abstract])

#3 #1 AND #2

Chinese databases were searched on China Knowledge Network (CNN), for example, with the search formula: SU % = 'osteoporosis' + 'bone mass' + 'bone density' + 'age-related bone loss' + 'osteoporotic fracture' AND SU % = 'clinical decision making' + 'decision aid' + 'decision support'. The timeframe for searching is from the establishment of the database to June 30, 2024.

### Screening of articles

We will use the reference management software EndNote X9 to download abstracts from each database to deduplicate search results. To ensure the consistency and reliability of the study selection process, all abstracts were independently reviewed by two different members of the research team. Any disagreements over inclusion were resolved through consensus and, where necessary, discussion with a third member of the review team. Following the abstract review, this process was replicated to complete the full-article review.

## Data extraction and synthesis

Two reviewers independently reviewed full-text and extracted data into Microsoft Excel. The data included general information such as authors, year of publication, country, study site, sample information, study type, intervention form, content elements, application scope, and outcome indicators. Narrative data synthesis was undertaken, and a meta-analysis was not deemed appropriate due to the nature of this review and the data included. Following the JBI methodology, we will conduct a pilot test of the extraction tool on three full-text articles to ensure its reliability, with results checked for consistency. The extraction of study design was executed with precision to ascertain the rigor and relevance of the studies, adhering to the PO and PICO levels of evidence delineated by the Centre for Evidence-Based Medicine (CEBM) for a nuanced understanding of the research methodologies employed.

## Results

### Study selection

We identified 1221 studies from electronic databases after duplicates were removed. We excluded 1047 on title and abstract screening. We excluded a further 156 on full-text screening for the following reasons: patient population ($n = 138$), study design ($n = 8$), no involvement of decision aids ($n = 7$), not available ($n = 3$). In total, 18 studies (all from published literature sources) were included (Fig 1).

### Basic characteristics of the included literature

Eighteen papers were studied in the United States (n = 7), the Netherlands (n = 3), Germany (n = 1), Italy (n = 1), Iceland (n = 1), Thailand (n = 1), India (n = 1), Israel (n = 1), the Philippines (n = 1), and Australia (n = 1). The types of studies included randomized controlled studies (n = 7), retrospective studies (n = 4), experimental-like studies (n = 3), mixed studies (n = 3), and qualitative studies (n = 1). The forms of interventions for DA included manuals (n = 6), website platforms (n = 6), videos (n = 2), and other multimedia interventions (n = 4). The basic characteristics of the included literature are shown in Table 1.

### Scope of application of decision support tools

The entire OP diagnosis and treatment process is involved in the usage of DA in osteoporosis patients. ① Diagnosis of OP (n = 5) [15–17,24,29]: Aids in subsequent decision-making by providing information on osteoporosis and individual fracture risk, as well as explaining future fracture risk in individuals who are left untreated (natural course). ② Making decisions about treatment (n = 7) [13,20,22–24,28,30]: Helps make decisions about prescription drugs, prevention of controllable risk factors, exercise, nutrition, and rehabilitation. ③ Suggested medication (n = 2) [18,19]: In order to promote medication

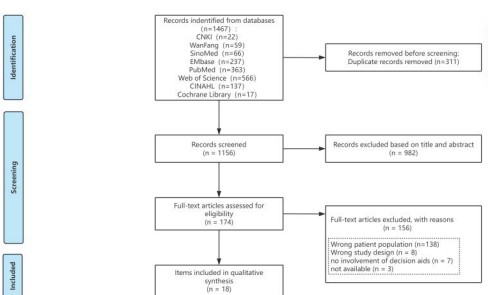

**Fig 1. Flow chart of literature screening.**

**Table 1. Basic characteristics of the included literature (n = 18).**

| Author | Year | Country | research site | study population | Study design | sample size | Forms of intervention | Content elements | Scope of Application | Out-come indicator |
|---|---|---|---|---|---|---|---|---|---|---|
| Lieke et al [13] | 2024 | the Netherlands | Hospital | Osteoporosis patients | hybridization study | 15 cases | brochure | ①③ | treatment | AC |
| Khanna et al [14] | 2023 | India | Hospital | Osteoporosis patients | A retrospective study | 1493 cases | network platform | ①②④ | Secondary prevention | AB |
| Al-Hashimi et al [15] | 2023 | German | Hospital | Patients with secondary osteoporosis | A retrospective study | 264 cases | brochure | ①② | diagnostic | ABC |
| Atiporn et al [16] | 2022 | Thailand | Hospital | Postmenopausal women with osteoporosis | A retrospective study | 407 cases | brochure | ①② | diagnostic | ABC |
| Cathleen et al [17] | 2022 | United States of America | Community and primary care clinics | Men with osteoporosis and primary care providers | A randomized controlled study | 176 cases | Brochures and electronic supplementary materials | ①③ | diagnostic | ABC |
| Bonaccorsi et al [18] | 2021 | Italy | research center | Menopausal women with osteoporosis | Type of experimental study | 2052 cases | network platform | ①②④ | treatment | ABC |
| Miller et al [19] | 2021 | United States of America | Hospital | Women 55 years and older with osteoporosis | A randomized controlled study | 1759 cases | network platform | ①④ | Treatment, secondary prevention | AB |
| Cornelissen et al [20] | 2021 | the Netherlands | Hospital | Osteoporosis patients | hybridization study | 248 cases | brochure | ①②③ | treatment | ABC |
| Goldshtein et al [21] | 2020 | Palestine | communal | Osteoporosis patients | Type of experimental study | 12329 cases | network platform | ①④ | treatment | ABC |
| Roxas et al [22] | 2020 | Philippine | Hospital | Osteoporosis patients and healthcare professionals | hybridization study | 12 cases | graphic flashcard | ①②③ | treatment | ABC |
| Olivo et al [23] | 2020 | United States of America | Hospital | Postmenopausal women with osteoporosis | A randomized controlled study | 225 cases | Brochure and accompanying video | ①③ | primary prevention | ABC |
| Gudmundsson et al [24] | 2019 | United States of America | Hospital | Osteoporosis patients | A retrospective study | 259 cases | Brochures and electronic supplementary materials | ①③ | diagnostic | B |
| Danila et al [25] | 2018 | United States of America | Hospital | Women with a self-reported history of fracture after age 45 who were not treated for osteoporosis | A randomized controlled study | 2684 cases | Video and Audio Narrative | ①③ | primary prevention | AB |
| Smallwood et al [26] | 2017 | United States of America | Primary care clinics | Post-menopausal women | A randomized controlled study | 50 cases | network platform | ①③④ | primary prevention | BC |
| Hiligsmann et al [27] | 2016 | the Netherlands | patients | Postmenopausal women with osteoporosis and health care workers | qualitative research | 12 cases | brochure | ①② | primary prevention | AB |
| Annie et al [28] | 2015 | Australia | Primary care clinics | Postmenopausal women with osteoporosis | A randomized controlled study | 79 cases | brochure | ①② | treatment | ABC |

*(Continued)*

Table 1. (Continued)

| Author | Year | Country | research site | study population | Study design | sample size | Forms of intervention | Content elements | Scope of Application | Out-come indicator |
|---|---|---|---|---|---|---|---|---|---|---|
| Halldorsson et al [29] | 2015 | Icelandic | Hospital | Osteoporosis patients | Type of experimental study | 308 cases | network platform | ①②④ | Diagnosis, secondary prevention, treatment | B |
| Montori et al [30] | 2011 | United States of America | Primary care Clinics | Postmenopausal women with osteoporosis | A randomized controlled study | 100 cases | Brochure and accompanying video | ①②③ | treatment | ABC |

Note: ① information guide; ② description of risk pros and cons associated with decision options; ③ clarification of values and care preferences; ④ interactive decision communication support.

Decision process indicator (A); Decision quality indicator (B); Feasibility evaluation (C).

readiness, medication trust, and medication adherence, as well as to support decision-making about clinical safety and efficacy, side effects, cost, frequency of administration, mode of administration, and route of administration, among other things. ④ Primary prevention of OP (n = 4) [21,25–27]: Choosing interventions for primary prevention. This includes health education, quitting smoking, improving one's diet, getting enough sunlight, exercising frequently, and taking calcium supplements. ⑤ Decisions about secondary prevention of OP (n = 3) [14,18,29]: Choosing a medicine and altering one's lifestyle.

## Content elements of decision support tools

The following are the primary components of DA. ① Informational advice [13–30]: Give detailed information about the patient's physical condition and the topic of decision-making, such as the course of osteoporosis, the patient's needs and priorities for making decisions at various stages of the disease, and the availability of information about osteoporosis. ② Outlining the risks and drawbacks of the options available for decision-making [14–16,19,20,23,27–30]: Help the patient or caregiver set a realistic expectation of benefits by outlining the risks and drawbacks of each choice related to the decision-making subject. ③ Values and care preferences are clarified [13,17,19,21,23–26,30]: questionnaires, intention analysis, and cognitive interviews are used to elicit values, care goals, treatment preferences, etc. ④ Interactive decision-making communication support [14,18,20,22,26,29]: assistance in making decisions with experts and peers. Through interfaces like chat rooms, discussion forums, and open comment sections, the website-based DA's user interaction feature enables users to exchange decision-making experiences and get peer support. Additionally, the DA can incorporate longitudinal data, like a patient's health status, into its proprietary EHR platform, where experts can offer decision-making support. Additionally, DA may include long-term information, such as patient's health status, into its special EHR platform, allowing experts to offer tailored guidance for decision-making.

## Outcome indicators for decision support tools

The outcome indicators pertain to three domains: feasibility assessment, decision quality indicators, and decision-making process indicators. ① Indicators of the decision-making process: The effect of DA on patient involvement in the decision-making process was documented in 15 research [13–23,25,27,28,30]. Of them, 13 studies [13–18,20–23,27,28,30] demonstrated that DA could lessen the financial burden on families and society, improve clinical decision-making and healthcare administration, improve patients' quality of life, and improve patients' perception of their risk of osteoporosis. The effects of DA on decision-makers adherence aspects were documented in three studies [18,19,28]. DA may enhance nurse-patient communication, facilitate communication and discussion with healthcare

professionals regarding treatment options and care goals, and improve patient or caregiver adherence in the decision-making process. However, two research [18,19]on the subject of medication recommendations demonstrated that it is challenging to put better medication regimens into practice in the actual world in order to increase drug adherence. Although more patients became consciously equipped for medication preventive therapy as a result of practice in the actual world, another study [28] revealed that the outcomes were not statistically significant. ② Indicators of decision quality: The effect of DA on the quality of decisions was documented in 17 investigations [14–30]. Among them, in terms of knowledge gain, 5 studies [21,23,25,28,30]demonstrated that DA boosted patients' or caregivers' knowledge connected to the decision topic. In terms of decision-making conflict, 3 studies [19,21,26] revealed that DA lowers decision-making conflict, including reducing uncertainty in the decision-making process and alleviating decision-making distress. Nevertheless, other research indicates that DA has little impact on conflict in decision-making [28]. According to Roxas et al. [23], patients may become more anxious and stop taking anti-osteoporosis drugs as a result of being informed of their side effects. Three studies [18,25,29] demonstrated that DA enhances the consistency of treatment plans and care objectives between patients or caregivers and medical personnel. Three studies [27,28,30] demonstrated that DA increased caregiver and patient satisfaction with care. While one study [22] that focused on identifying osteoporosis risk did not show a change in this outcome indicator, two investigations [14,17] of DA-assisted fracture risk assessment found lower DXA and fracture expenses, which reduced the financial strain on families and society as a whole. The impact of DA on patients' quality of life was documented in three investigations [15,19,28], but none of the differences were statistically significant. ③ Feasibility indicators: DA feasibility indicators were the subject of 12 investigations [13,15–17,19–23,26,28,30]. One study [22] on patient treatment decision-making noted that DA was more formally oriented towards patient counseling and lacked a more direct and comprehensive discussion of treatment options with healthcare professionals. Ten of the studies [13,15–17,19–21,26,28,30] demonstrated that DA performed well in terms of acceptability, clarity of decision-making topics, clarity of content, comprehensibility, helpfulness in care planning for people with osteoporosis or carers, and recommendability. According to a different study [23], patients' values did not align with the themes of decision-making, and choice regret led to decreased quality of life, adherence, and satisfaction.

## Discussion

### To better serve patients' decision-making needs, the use of decision aids in osteoporosis patients needs to be increased

The findings of this study demonstrated that the use of decision aids in OP patients was concentrated on the disease's treatment [13,18–20,22,23,28–30], including medication selection and treatment options; however, there was comparatively little use of decision aids in OP diagnosis, primary prevention, and secondary prevention. On the other hand, the content parts of DA primarily represent the comprehensive and personalized decision-making demands of patients. Information guidance, risk explanation, pros and cons of choices for decision-making, definition of values and care goals, and interactive decision-making communication support are the four components of DA, which is more thorough. Clarifying values and care preferences, as well as offering information and assistance, were two or more content aspects found in all included research [13–30]. Simultaneously, the DA content parts were tailored for several application areas. Understanding the benefits and drawbacks of various treatment alternatives, creating detailed action plans for patients, and offering support resources to families are all further components of DA for treatment plan decision-making [13,19,29]. According to a study by Cathleen et al. [17], knowing the causes of behavioral and lifestyle changes is crucial and can increase patient compliance, according to both patients and medical professionals. In chronic disease populations, DAs can serve as decision aids for dietary management, physical activity management, and lifestyle modification, according to a number of studies [9,31]. This makes it possible to swiftly and precisely create clinically tailored and improved dietary, exercise, and lifestyle intervention programs. Consequently, there is a strong chance that decision aids will be used to prevent OP. It is advised that decision aids be used in health education and other nursing procedures to encourage patients to change

their lifestyles and behaviors, as well as that the viability and application of decision aids in the primary and secondary prevention of OP patients be thoroughly investigated.

## It is still necessary to investigate how well decision aids work for people with osteoporosis

Most patients gave decision aids a positive rating, indicating that they had good feasibility and beneficial impacts [13,17,23,27,30]. ① Impact on the decision-making process: the DA optimizes the decision-making process, improves the quality of patient communication [23,27], and increases the perception of risks and rewards [14,17,18,22,27,30]. A number of factors, including the confounding trend of adherence with the use of DA, the various intervention environments, the attitudes of healthcare professionals about DA, and other factors, may contribute to the contentious conclusion on the influence on adherence [19,28]. Future research must increase the findings' application in actual clinical settings and conduct pertinent investigations depending on the national situation. ② Affects on the quality of decision-making: DA can improve satisfaction, raise patients' or their caregivers' understanding of decision-making themes, and improve patients' and healthcare professionals' agreement on treatment plans and care goals [2,18,21,23,25]. However, there was inconsistency in the effects on decision-making conflict, family and socioeconomic burden, and quality of life [15,17,21,22,26,28,29]. This could be due to the type of study, the patient's disease stage, the decision-making topics, and the intervention period. To ensure more scientific results, future research must conduct randomized controlled studies with large sample sizes. ③ Feasibility evaluation: Most patients gave the DA high ratings for acceptability, comprehensibility, and clarity of content presentation [13,15–17,19–21,26,28,30], and they said it was helpful for planning their current or future care [21,26]. While precise risk identification, individualized decision-making options, and professional support are beneficial, some participants felt that DA had certain limitations, such as a lack of knowledge about various decision-making options, irrelevant decision-making content, and a lack of alignment with the decision maker's values [22,23,24]. It is advantageous to encourage the use and spread of DA in the area of clinical decision-making for patients with osteoporosis by examining the obstacles and enablers to its application and investigating ways to improve it.

## Improvement of professional assistance to enable medical personnel to administer tailored interventions for patients

In addition to producing individualized patient reports, decision aids have been proposed as a successful way to educate patients and medical professionals on best practices [29]. When used effectively, DAs can raise awareness of osteoporosis among the general public and medical professionals, ensuring that people with the condition receive the best care and have the best possible prognosis [32]. Even for non-specialized clinical health experts, individuals actually struggle to understand the osteoporosis and related fracture risks that these systems present [33]. ① Due to their inadequate health literacy, older, less educated patients use decision aids less frequently. On the other hand, older OP patients are a crucial demographic to concentrate on because they are more likely to be older and have lower educational attainment. It is advised that more studies be done on shared decision-making for senior OP patients and that practical ways to help senior ACS patients take part in decision-making be thoroughly investigated. ② Patient acceptance is greatly impacted by the timing of the decision-making support that medical practitioners offer. Both patients and medical professionals prefer that the doctor make the decision, particularly when the patient is in the emergency phase of the illness [13,23,25]. This might have to do with the fact that patients don't have enough time to weigh their options, get information, and think about alternate possibilities in life-threatening emergencies [34]. Nonetheless, patients continue to expect their doctors to consider their opinions, indicating that acute phase patients still need to communicate their preferences and expectations [13]. In the emergency situation, DA can help patients receive information and save time when making decisions. It can also help with shared decision-making implementation and enable conversations about shared decision-making [35]. To help healthcare professionals better understand patient

preferences and expectations in high-stress clinical settings, more research on shared decision making for emergency situations is advised. This research should focus on DAs appropriate for emergency medical decision making. ③ Patients who lack decision-related knowledge may find it more challenging to make decisions because they are unable to form an opinion [21]. Trained choice coaches who assist patients or their families in making decisions are referred to as decision coaches. When combined with evidence-based care, decision coaching can help patients become more knowledgeable about making decisions [36]. The country can learn from this model, expand the professional role of nurses, and give patients evidence-based decision-making guidance to improve the type of decision-making support for patients. In foreign studies, decision coaches are typically nurses who have undergone systematic training and possess the necessary qualifications [37].

### Decision aids are used in a variety of forms, and web-based decision aids have more room for development

Decision aid manuals, films, website platforms, and other multimedia formats are examples of current DA uses for clinical decision making in patients with osteoporosis. Web-based DA has advanced quickly in recent years to help people with osteoporosis make decisions, although its use is still a little lacking when compared to classic decision-assistance manuals. As a result, DAs distributed through web-based platforms and their derivative forms continue to have a wide range of potential uses among osteoporosis patients in the future. The web-based DA can offer users more individualized decision-making solutions, a wider range of decision assistance options, and richer information resources than the paper version and audio/video application forms. Furthermore, the paternalistic paradigm of healthcare decision-making has given way to a shared one, and the main benefit of web-based DA is that the latter can offer interactive decision communication help. A decision assistance platform was created by Smallwood et al. [26] that offers a dialogue box for professionals to communicate with. This allows users to ask for advice and discuss their confusion during the decision-making process with professionals in real time. Healthcare providers are also able to promptly adapt to their patients' demands. According to the study [23], interactive communication can help patients and healthcare professionals. It can help patients with varying educational backgrounds understand disease knowledge and lessen regret about their decisions; it can improve the ability to discuss complex decision-making issues and obtain professional information; and it can give healthcare professionals evidence-based information and key points. However, a well-designed navigation system and a user-friendly, low-threshold access program are essential for web-based DA programs to be effective. Overly specialized language searches, complicated page designs, and program constraints limited to computer-based access have been found to cause difficulties in use and a rise in decision-making conflicts for older users with poorer information literacy skills, according to some research [21,29]. In the future, when developing a web-based decision aid for patients with osteoporosis, more thought should be given to patient preferences and individual learning styles to increase the clinical generalizability of the tool. For example, the application version of the DA could be expanded to make it easier to use; the layout of the decision aid page could be optimized to increase the scope of its application; and patients could find the information they need to make decisions more easily.

### Future research

A comprehensive assessment of decision aids for osteoporosis patients was used in this study to assess how well they manage the condition, decrease fragility fractures, and improve bone density tests and osteoporosis therapy; however, further research is required in the following areas. ① Enhance the content of decision support for osteoporosis treatment and offer decision support for refined lifestyle intervention: individualized and improved nutrition, exercise, and rehabilitation intervention programs are significant factors influencing the efficacy of DA application, and the study's decision aids were primarily focused on treatment choices and medication recommendations, with no choices for lifestyle intervention [13,18–20,22–24,28,30]. The content of OP treatment decision-making can be further enhanced in the future, and efforts can be made to create a DA for lifestyle interventions related to osteoporosis in

 

order to give patients with the disease more individualized lifestyle advice. ② Standardize the DA evaluation indexes for osteoporosis and diversify the DA evaluation indexes for clinical application effects: In China, the DA evaluation indexes for osteoporosis are primarily based on patient clinical outcomes, with less attention paid to DA's usability, user experience, and health economics assessment. Future research should focus on evaluating the health economic effects of DAs in the field of osteoporosis in order to provide a healthy economic basis for the health management of patients with osteoporosis. Researchers should also test DAs for ease of use and comprehensibility before formally implementing them in the clinic or when designing them. This will help DAs better integrate with the clinical work environment and processes. In the meanwhile, randomized controlled trials with sizable sample sizes must be conducted in the future with the goal of guaranteeing uniform interventions for the variables that currently yield contradictory results, such as decision conflict and adherence. ③ Enhance DA's professional support and decision-support options by forming a clinical decision support team headed by advanced practice nurses, forming a multidisciplinary collaborative team, and hosting organized joint nursing conferences. ④ Using cutting-edge international experience, create decision-supporting tools tailored to China's unique circumstances. Currently, China uses fewer related instruments than Europe and the United States when it comes to clinical decision-making for patients with osteoporosis. Since China, Europe, and the United States have different medical environments, cultural backgrounds, and osteoporosis diagnosis and treatment systems, the creation or localization of DA must be based on the unique clinical practice situation in China. Its use is complicated and decision-making disputes are increased because domestic decision-making in OP patient-related elements depends increasingly on the involvement of healthcare professionals. Therefore, in order to improve DA's adaption to the home clinical setting, the creation of localized decision aids must also consider the desires and preferences of patients and their families.

## Limitations

Our scoping review has some limitations. First, this scoping review only included studies completed in English, so relevant studies in other languages may have been excluded. Lastly, like all literature reviews, our results are limited by the published research available. With more data available, the results might change. Second, the review was restricted to research that used DXA data, which meant that studies from other imaging modalities (such as magnetic resonance imaging) that would have offered more information were not included. Finally, in keeping with the scoping review framework (which aims to identify and map the scope of available evidence rather than the quality of evidence identified), we did not assess the quality of the included studies. Therefore, it was unclear if discrepancies in some results were due to variations in the quality of the methods used in different investigations.

## Conclusion

This study summarizes the research on the application of DA in the field of clinical decision-making in osteoporosis patients, which has a good prospect of application in OP patients, with the help of DA to increase the patient's understanding of risk information and encourage them to express their decision-making preferences, which can help to improve the decision-making experience and the quality of decision-making. The effectiveness and feasibility of decision aids have been preliminarily confirmed, but there are still some problems in the application of DA in osteoporosis patients, such as the scope of application needs to be further explored, fewer types of DA, and lack of comprehensiveness of assessment tools. In the future, it is necessary to further explore the application effect and implementation method of decision aids in OP patients, to strengthen the professional support in the decision-making process, and to promote patient-centered care in OP diagnosis and treatment. At the same time, the development of localized decision aids in China should be accelerated to improve the quality of decision aids and reduce the decision pressure of osteoporosis patients in the clinical decision-making process.

## Supporting information

**S1 File. PRISMA-ScR Checklist.**
(DOCX)

**S2 File. Data extraction phase.**
(DOCX)

**S3 File. Summary table of included studies.**
(XLSX)

**S4 File. The data extraction table template.**
(DOCX)

**S5 File. The search strategy for PubMed.**
(DOCX)

## Author contributions

**Data curation:** Yanyu Fang, Siqi Li.

**Formal analysis:** Yanyu Fang, Qin Jia.

**Funding acquisition:** yaqin Dai.

**Methodology:** Yanyu Fang, yaqin Dai, Siqi Li.

**Supervision:** Qin Jia, yaqin Dai.

**Writing – original draft:** Yanyu Fang.

**Writing – review & editing:** Qin Jia.

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
