## [Decision Letter · Decision Letter 0]

Dear Dr. Jia,

Thank you for submitting your manuscript to PLOS ONE. After careful consideration, we feel that it has merit but does not fully meet PLOS ONE’s publication criteria as it currently stands. Therefore, we invite you to submit a revised version of the manuscript that addresses the points raised during the review process.

We look forward to receiving your revised manuscript.

Kind regards,

Chung-Ta Chang

Academic Editor

PLOS ONE

 [This work was supported by the Zhejiang Provincial Department of Science and Technology Basic Public Welfare Research Funding Project (Grant number LTGY23H170005); the General project of Zhejiang Provincial Medical and Health Science and Technology Program (Grant number 2024KY644).].

5. Please amend the manuscript submission data (via Edit Submission) to include author Dai Yaqin, Li Siqi.

6. As required by our policy on Data Availability, please ensure your manuscript or supplementary information includes the following:

Additional Editor Comments (if provided):

Reviewers' comments:

Reviewer's Responses to Questions

**Comments to the Author**

1. Is the manuscript technically sound, and do the data support the conclusions?

Reviewer #1: Yes

Reviewer #2: Yes

Reviewer #3: Yes

2. Has the statistical analysis been performed appropriately and rigorously?

Reviewer #1: Yes

Reviewer #2: N/A

Reviewer #3: N/A

3. Have the authors made all data underlying the findings in their manuscript fully available?

Reviewer #1: Yes

Reviewer #2: Yes

Reviewer #3: Yes

4. Is the manuscript presented in an intelligible fashion and written in standard English?

Reviewer #1: Yes

Reviewer #2: Yes

Reviewer #3: Yes

Reviewer #1: This scoping review aims to describe the use and effectiveness of decision aids (DAs) in clinical decision-making for individuals with osteoporosis. The authors conducted a systematic search across multiple databases to identify relevant literature and synthesized the findings. The topic is relevant to improving patient care and shared decision-making in osteoporosis management.

Recommendations:

Methods: As mentioned earlier, adding a brief description of any approach used to ensure the rigor of the data synthesis would be beneficial.

Discussion: While the authors mention future research directions, expanding on the practical implications of implementing decision aids in various clinical settings would further strengthen the discussion. For example, what are the barriers and facilitators to the adoption of decision aids in osteoporosis care?

Minor Editing: A thorough review for minor grammatical or typographical errors is recommended.

Reviewer #2: -In the abstract, the methods need to be clarified and mention the criteria you used to select the studies

-In the abstract, the results need to be clarified and mention the no. of the studies that met your criteria

-The rationale and significance of the study need to be clarified.

-It is better to mention in the results the quality of the studies used in the review if it is available

-The reference cited on no. 8 need to be revised as its year is 2017 not 2024.

-Reconsider the type of the article mentioned as it is a review article not a research article

-Revise the citation style (References) used to be commitment with the submission guidelines of the journal

Reviewer #3: 1. !This According to studies [9], DA can improve the accuracy of risk assessment, decrease decision-making conflicts, raise patient understanding and decision-making engagement, and boost the adoption of preventive interventions”. Here please kindly cite all the studies you refer to or rephrase.

2. Since this is a Scoping Review, the objective must be defined at the end of the introduction. Please start by clearly defining the purpose and scope of your scoping review.

3. For the selection and removal of duplicates, the authors used a Reference Management Tool such as EndNote? If so, please specify.

4. In the inclusion and exclusion criteria part it is recommended, but not mandatory, that authors specify what was the range of years for the selection of articles?

**Do you want your identity to be public for this peer review?** For information about this choice, including consent withdrawal, please see our Privacy Policy

Reviewer #1: No

Reviewer #2: No

Reviewer #3: No

---

## [Author Response · Author response to Decision Letter 1]

19 Jun 2025

Dear Editor,

Thank you very much for giving us an opportunity to revise our manuscript. We appreciate the editor and reviewers sincerely for their constructive comments and suggestions to our manuscript entitled “Decision Aids in Patients with Osteoporosis: A Scoping Review” (NO: PONE-D-25-16775).

Those comments are all valuable and very helpful for revising and improving our Paper, as well as the important guiding significance to our following research. We have presented detailed response to reviewers’ comments. According to the reviewers’ comments, we have made a careful revision on the original manuscript. All changes in the manuscript are highlighted in yellow for easy reference (or tracked changes if applicable).

Below, we provide a point-by-point response to all comments.

Responses to Editor’s Comments

1. Please ensure that your manuscript meets PLOS ONE's style requirements, including those for file naming. Please provide an amended statement that declares *all* the funding or sources of support (whether external or internal to your organization) received during this study.

Response: Thank you. We have carefully read the PLOS One manuscript style template and carefully revised and improved it.

2. Please provide an amended statement that declares *all* the funding or sources of support (whether external or internal to your organization) received during this study. Please also include the statement “There was no additional external funding received for this study.” in your updated Funding Statement.

Response: Thank you. We have updated the Fund Statement. “This study was supported by the Basic Public Welfare Research Program of Zhejiang Province (Grant No. LTGY23H170005) and Zhejiang Medical and Health Science and Technology Project (Grant No. 2024KY644). There was no additional external funding received for this study.”

3. Please confirm at this time whether or not your submission contains all raw data required to replicate the results of your study.

Response: Thank you. No datasets were generated or analysed during the current study. All relevant data from this study will be made available upon study completion.

4. PLOS requires an ORCID iD for the corresponding author in Editorial Manager on papers submitted after December 6th, 2016. Please ensure that you have an ORCID iD and that it is validated in Editorial Manager.

Response: Thank you. We have it and it has been verified.

5. Please amend the manuscript submission data (via Edit Submission) to include author Dai Yaqin, Li Siqi.

Response: Thank you. We have modified this as suggested. Please refer to the manuscript data in the submission system.

6. As required by our policy on Data Availability, please ensure your manuscript or supplementary information includes the following: A numbered table of all studies identified in the literature search, including those that were excluded from the analyses. For every excluded study, the table should list the reason(s) for exclusion.

Response: Thank you. We've added files to support the Data Availability, including a summary table of included studies, the data extraction table template, the data extraction phase and the search strategy for PubMed.

7. Please review your reference list to ensure that it is complete and correct.

Response: Thank you. We have thoroughly rechecked all references in the manuscript and confirm that: All citations are accurately matched with the reference list; Reference formatting strictly follows the journal's guidelines; No redundant or omitted references were found; Each entry includes complete metadata.

Responses to reviewer #1:

1. In the methods, adding a brief description of any approach used to ensure the rigor of the data synthesis would be beneficial.

Response: Thank you for pointing out this defect in our manuscript. According to the reviewer’s suggestions, we carefully extracted relevant data from the included studies using a methodological framework provided by the Joanna Briggs Institute (JBI). The extraction of the study design is performed precisely to determine the rigor and relevance of the study, and we strictly adhere to the PO and PICO evidence levels delineated by the Center for Evidence-Based Medicine (CEBM) to provide a nuanced understanding of the research methods employed. (Data extraction and synthesis, line 6)

2. In the discussion, while the authors mention future research directions, expanding on the practical implications of implementing decision aids in various clinical settings would further strengthen the discussion. For example, what are the barriers and facilitators to the adoption of decision aids in osteoporosis care?

Response: We appreciate the reviewer's insightful comment regarding the clinical significance of decision aids. As we discussed in the second point of our Discussion section (Discussion, Paragraph 2), current evidence on the effectiveness of decision aids remains controversial. While our study identified some barriers and facilitators for implementing decision aids in osteoporosis care based on literature synthesis, we fully agree that these factors warrant more systematic investigation. We have now explicitly stated this as a key direction for future research in the revised manuscript.

3. A thorough review for minor grammatical or typographical errors is recommended.

Response: Thank you for pointing out this defect in our manuscript. According to the reviewer’s suggestions, we have thoroughly proofread the entire manuscript and corrected all grammatical inaccuracies and typographical errors. All modifications are highlighted in the revised manuscript for easy reference.

Responses to reviewer #2:

1. In the abstract, the methods need to be clarified and mention the criteria you used to select the studies. The results need to be clarified and mention the no. of the studies that met your criteria.

Response: Thank you for pointing out this defect in our manuscript. According to the reviewer’s suggestions, we have meticulously refined the Methods and Results sections. Additionally, upon further literature review, we have enhanced the Abstract by incorporating the study’s rationale and detailed inclusion criteria. (Abstract, Paragraphs 1 and 3)

2. The rationale and significance of the study need to be clarified.

Response: Thank you for pointing out this defect in our manuscript. In the Abstract (Line 1-5), we added a dedicated paragraph outlining the rationale of the study: Although various methods exist for osteoporosis prevention, most patients fail to receive optimal treatment due to information asymmetry between physicians and patients, as well as limited consultation time. Current literature (Paskins Z et al., 2020; Nogués X et al., 2022) suggests that decision aids can support clinical decision-making by improving patients' risk perception and treatment acceptance. In the Discussion (Paragraph 2), we discussed the use of decision aids in osteoporosis patients, which have been demonstrated to have positive outcomes via integration; nevertheless, the findings are now contentious for several indicators, such as adherence and choice conflicts. In the Future research (Line 1-3), we explicitly stated the clinical implications, A comprehensive assessment of decision aids for osteoporosis patients was used in this study to assess how well they manage the condition, decrease fragility fractures, and improve bone density tests and osteoporosis therapy.

3. It is better to mention in the results the quality of the studies used in the review if it is available.

Response: Thank you for pointing out this defect in our manuscript. Although scoping reviews typically do not involve detailed assessments of study quality, we will incorporate standardized quality assessment tools in subsequent research to rigorously evaluate included studies and mitigate data extraction bias.

4. The reference cited on no. 8 need to be revised as its year is 2017 not 2024.

Response: Thank you for pointing out this defect in our manuscript. After rechecking the original source (STACEY D et al., 2017), we confirm the correct publication year is 2017 and have revised all relevant citations in the Introduction, Paragraph 2.

5. Reconsider the type of the article mentioned as it is a review article not a research article.

Response: We thank the reviewer for highlighting this issue. This submission is indeed a scoping review, but we have followed editor’s suggestion and selected "Research Article" as the submission type, as the journal's online system does not currently provide a specific category for review articles.

6. Revise the citation style (References) used to be commitment with the submission guidelines of the journal.

Response: Thank you for pointing out this defect in our manuscript. We have now carefully revised all references throughout the manuscript to strictly comply with the journal's style guidelines. All in-text citations and reference list entries have been reformatted according to PLOS One's “Vancouver” style. References were generated using the PLOS One template in EndNote X9, with manual cross-checking against original sources.

Responses to reviewer #3:

1. !This According to studies [9], DA can improve the accuracy of risk assessment, decrease decision-making conflicts, raise patient understanding and decision-making engagement, and boost the adoption of preventive interventions”. Here please kindly cite all the studies you refer to or rephrase.

Response: Thank you for pointing out this defect in our manuscript. According to the reviewer’ suggestions, we have now cited all supporting references for each claimed benefit of DAs: Improved risk assessment accuracy:[5]; Reduced decisional conflict:[9]; Enhanced patient understanding:[2]; Increased preventive intervention uptake:[6](Abstract, Line 19-22). In the manuscript, it has been modified to “According to studies, DA can improve the accuracy of risk assessment[5], decrease decision-making conflicts[9], raise patient understanding and decision-making engagement[2], and boost the adoption of preventive interventions[6]”.

2. Since this is a Scoping Review, the objective must be defined at the end of the introduction. Please start by clearly defining the purpose and scope of your scoping review.

Response: Thank you for pointing out this defect in our manuscript. According to the reviewer’s suggestions, we have restructured the introduction's final paragraph to explicitly state the scoping review's purpose and scope (Paragraph 3, Lines 5-9). In the manuscript, it has been modified to “In summary, this scoping review systematically synthesizes decision topics, content elements, outcome types, and effectiveness of decision aids (DAs) in clinical decision-making for osteoporosis patients. By mapping existing evidence, we identify current knowledge gaps and provide actionable recommendations to guide future research directions in this field”.

3. For the selection and removal of duplicates, the authors used a Reference Management Tool such as EndNote? If so, please specify.

Response: We appreciate the reviewer's attention to methodological rigor. As detailed in the revised Methods section (Paragraph 6, Lines 1-3): Primary deduplication was performed using EndNote X9 (Clarivate Analytics) with the following steps, automated duplicate detection (threshold: ≥90% similarity in title/author/year), manual verification of all potential duplicates.

4. In the inclusion and exclusion criteria part it is recommended, but not mandatory, that authors specify what was the range of years for the selection of articles?

Response: We thank the reviewer for highlighting this issue. In our manuscript, we specified the search timeframe as “from the establishment of the database to June 30, 2024” to ensure comprehensive coverage of all potentially relevant studies. This approach was taken to avoid excluding any seminal early works that might have been overlooked with an arbitrary start date cutoff. We acknowledge the reviewer's valuable perspective and will systematically evaluate the potential impact of publication year ranges on our findings in future analyses.

Sincerely,

Yanyu Fang

Zhejiang Chinese Medical University

Email: 3496946732@qq.com

---

## [Editor Report · Decision Letter 1]

Decision aids in patients with osteoporosis: A scoping review

PONE-D-25-16775R1

Dear Dr. Jia,

We’re pleased to inform you that your manuscript has been judged scientifically suitable for publication and will be formally accepted for publication once it meets all outstanding technical requirements.

Kind regards,

Chung-Ta Chang

Academic Editor

PLOS ONE
---

## [Editor Report · Acceptance letter]

PONE-D-25-16775R1

PLOS ONE

Dear Dr. Jia,

I'm pleased to inform you that your manuscript has been deemed suitable for publication in PLOS ONE. Congratulations! Your manuscript is now being handed over to our production team.

Kind regards,

on behalf of

Dr. Chung-Ta Chang

Academic Editor

PLOS ONE